# Using Wastewater Surveillance to Compare COVID-19 Outbreaks during the Easter Holidays over a 2-Year Period in Cape Town, South Africa

**DOI:** 10.3390/v15010162

**Published:** 2023-01-05

**Authors:** Nomfundo Mahlangeni, Renée Street, Suranie Horn, Angela Mathee, Noluxabiso Mangwana, Stephanie Dias, Jyoti Rajan Sharma, Pritika Ramharack, Johan Louw, Tarylee Reddy, Swastika Surujlal-Naicker, Sizwe Nkambule, Candice Webster, Mongezi Mdhluli, Glenda Gray, Christo Muller, Rabia Johnson

**Affiliations:** 1Environment & Health Research Unit, South African Medical Research Council (SAMRC), Johannesburg 2028, South Africa; 2Environmental Health Department, Faculty of Health Sciences, University of Johannesburg, Johannesburg 2028, South Africa; 3Occupational Hygiene and Health Research Initiative, North-West University, Potchefstroom 2531, South Africa; 4Biomedical Research and Innovation Platform (BRIP), South African Medical Research Council (SAMRC), Tygerberg 7505, South Africa; 5Department of Microbiology, Stellenbosch University, Stellenbosch 7600, South Africa; 6Discipline of Pharmaceutical Sciences, School of Health Sciences, University of KwaZulu-Natal, Westville Campus, Durban 4001, South Africa; 7Department of Biochemistry and Microbiology, University of Zululand, KwaDlangezwa 3886, South Africa; 8Biostatistics Research Unit, South African Medical Research Council (SAMRC), Durban 4091, South Africa; 9Scientific Services, Water and Sanitation Department, City of Cape Town Metropolitan Municipality, Cape Town 8000, South Africa; 10Chief Research Operations Office, South African Medical Research Council (SAMRC), Tygerberg 7050, South Africa; 11Office of the President, South African Medical Research Council (SAMRC), Tygerberg 7050, South Africa; 12Division of Medical Physiology, Faculty of Medicine and Health Sciences, Centre for Cardio-Metabolic Research in Africa, Stellenbosch University, Stellenbosch 7600, South Africa

**Keywords:** SARS-CoV-2, wastewater surveillance, Easter holidays, wastewater treatment plants

## Abstract

Wastewater surveillance of severe acute respiratory syndrome coronavirus 2 (SARS-CoV-2) has shown to be an important approach to determine early outbreaks of infections. Wastewater-based epidemiology (WBE) is regarded as a complementary tool for monitoring SARS-CoV-2 trends in communities. In this study, the changes in the SARS-CoV-2 RNA levels in wastewater during Easter holidays in 2021 and 2022 in the City of Cape Town were monitored over nine weeks. Our findings showed a statistically significant difference in the SARS-CoV-2 RNA viral load between the study weeks over the Easter period in 2021 and 2022, except for study week 1 and 4. During the Easter week, 52% of the wastewater treatment plants moved from the lower (low viral RNA) category in 2021 to the higher (medium to very high viral RNA) categories in 2022. As a result, the median SARS-CoV-2 viral loads where higher during the Easter week in 2022 than Easter week in 2021 (*p* = 0.0052). Mixed-effects model showed an association between the SARS-CoV-2 RNA viral loads and Easter week over the Easter period in 2021 only (*p* < 0.01). The study highlights the potential of WBE to track outbreaks during the holiday period.

## 1. Introduction

In response to the coronavirus 2019 (COVID-19) pandemic, many countries have employed wastewater surveillance of severe acute respiratory syndrome coronavirus 2 (SARS-CoV-2) as an additional approach to tracking the virus in the population. Unlike clinical testing which is limited to testing one individual at a time, a pooled sample collected from the wastewater treatment facility (serving a catchment area) is tested for the presence of inactivated SARS-CoV-2. The pooled samples can provide us with information about the circulation of the virus in that community. Wastewater surveillance of SARS-CoV-2 has shown to be an important tool at a time where emerging variants of concern are circulating in the population. Several researchers have tracked SARS-CoV-2 variants in wastewater [1,2,3,4,5,6]. 

From the first report of the detection of SARS-CoV-2 RNA in wastewater, many countries have designed and implemented SARS-CoV-2 wastewater surveillance programs to mitigate the impact of the virus in communities [7,8,9,10,11,12,13,14]. The SARS-CoV-2 RNA in wastewater can be detected at low levels when prevalence is low, and the presence of the virus can be detected amongst asymptomatic individuals [15,16,17]. Holiday gathering and traveling have played a significant role in the increased transmission of COVID-19 in communities [18,19]. Previous research confirmed the connection between human mobility and the spread of the SARS-CoV-2 virus [20,21]. In early 2021 many countries lifted their COVID-19 travel restrictions to restore the travel and tourism sector. The easing down of COVID-19 regulations worldwide has resulted in symptomatic individuals not testing, increased gatherings and mobility. This also increased the probability of the importation of new COVID-19 variant strains into countries. Wastewater surveillance data can be used by health authorities to monitor community trends and as needed, can encourage the public to take preventive measures to decrease the spread of COVID-19.

The City of Cape Town metropolitan municipality is one of the leading COVID-19 hotspots in the Western Cape province, South Africa. The City of Cape Town metro is home to 4.5 million people and is a world-famous tourism hub. The Western Cape province reported a total of 702,983 confirmed COVID-19 cases and 22,388 deaths to date. South Africa had several public holidays (National holiday on 21 March and a series of Easter-related holiday periods observed from Good Friday to Easter Monday). This period also falls into the end of the school term and for these reasons travelling increases during this time. In addition to travel, many individuals often gather at large social and religious events over the observed Easter period. Public health experts and national governments warned that travelling, social and religious gatherings during the holiday period may result in increased COVID-19 infections [22,23,24]. In South Africa, about 0.5% and 31% of the population was fully vaccinated during the Easter holidays in 2021 and in 2022, respectively [25]. In this study, we evaluate the trends in SARS-CoV-2 RNA viral load in wastewater during the Easter period in 2021 and Easter period in 2022 in the City of Cape Town. 

## 2. Materials and Methods

### 2.1. Wastewater Sample Collection and Processing

Grab wastewater samples were collected once a week on a Monday from the inlet of the 21 wastewater treatment plants (WWTPs) in the City of Cape Town during the Easter period of 2021 and Easter 2022, for nine weeks per year (Appendix A). The characteristics of the WWTPs are provided in Appendix A. About 500 mL of the influent sample was collected and transported on ice to the laboratory for analysis. 

In the laboratory, the total RNA was extracted with the Qiagen RNeasy^®^ PowerSoil^®^ Kit, as per the manufacturer′s instructions (Qiagen, Hilden, Germany) following the method described earlier [26,27]. In brief, 100 mL of influent wastewater was centrifuged at 2500× *g* for 20 min after which 2.5 mL of the pellet was used for the analysis. The quality of the total RNA was determined with a NanoDrop^®^ ND-1000 instrument (Nanodrop Technologies, Wilmington, NC, USA). A clinical SARS-CoV-2 nasopharyngeal swab sample with known viral copies was spiked into the wastewater sample. The extraction method was previously tested for efficiency [28].

### 2.2. SARS-CoV-2 Quantification by RT-qPCR

A one-step quantitative reverse transcription-polymerase chain reaction (RT-qPCR) assay was used to quantify SARS-CoV-2 nucleocapsid (N1 and N2) primer sets (Whitehead Scientific, Integrated DNA Technologies, Coralville, IA, USA) (Appendix A) [29]. The analysis was conducted using the one-step RT-qPCR reaction kit (iTaqTM Universal Probes One-Step Reaction kit, Bio-Rad Laboratories, Richmond, CA, USA) as described by Johnson et al. [3]. The SARS-CoV-2 RNA concentration was estimated on the QuantStudio™ 7 Flex Real-Time PCR System (ABI instrument, Life Technologies, Carlsbad, CA, USA). The cycling conditions were as follows: 50 °C for 10 min and 95 °C for 3 min, followed by 40 cycles consisting of 95 °C for 15 s and 60 °C for 60 s. All reactions were performed in duplicates and a template control was included for each experimental run. 

### 2.3. Data Analysis

The COVID-19 confirmed cases and percent PCR proportion positive (PCR confirmed cases/PCR tests done × 100%) were obtained from the Western Cape COVID-19 public-facing dashboard [30]. The confirmed cases and positivity rates were presented as seven-day moving averages. The data for the population served by the WWTP and design capacities were obtained from City of Cape Town metropolitan municipality. To account for the variations in the capacities of the WWTPs and the number of inhabitants in the catchment, the SARS-CoV-2 RNA signal was reported as gene copies/day/100,000 inhabitants. The population normalized viral load was calculated using Equation (1) by Gonzalez et al. [31] with some modifications.
(1)$$L_{\mathrm{WWTP}}=\frac{C_{\mathrm{WWTP}}\;\times \;V^{*}\;\times \;1\;\times \;10^{9}}{P}\;\times \;100,000$$
where L_WWTP_ is the population normalized SARS-CoV-2 viral loads in wastewater of each WWTP (genome copies per day per 100,000 inhabitants), C_WWTP_ is the SARS-CoV-2 RNA concentration (genome copies/mL), V* is the capacity of the WWTP (mL/day), and P is the population served. As daily inflow of each of the WWTPs was not available, the WWTP capacity was used as a substitute. The N1 and N2 assays used to detect viral copy number were used for the calculation. SARS-CoV-2 viral load below the limit of detection (LOD) (700 genome copies/mL) were replaced by half of LOD.

Due to the changes in the Easter dates each year, the Easter period in 2021 and Easter period in 2022 were placed according to the study week. The Easter week represented the Easter Monday. Study week 4 consisted of a national holiday, 21 March 2021 and 22 March 2022 (21 March fall on a Sunday therefore the Monday is recognized as a public holiday). The normality of variables was tested with the Shapiro–Wilk test. Wilcoxon signed rank test (non-parametric test) was performed to evaluate the mean differences between SARS-CoV-2 RNA viral load in wastewater in Easter period 2021 and 2022. Mixed-effects models was used to examine the effects of Easter week on the SARS-CoV-2 viral load in wastewater. All figures were plotted using the R version 4.2.1 with packages dplyr, purr and ggplot2. 

### 2.4. Spatial Data 

Suburb shapefiles were obtained from the City of Cape Town′s open data portal. Coordinates for each WWTP were collected using a handheld GPS and verified using Google Earth. All maps were produces using ArcGIS 10.6.1 (ESRI, Durban, South Africa).

## 3. Results

A total of 354 influent samples were collected from 21 WWTPs on the Easter holidays 2021 and Easter holidays 2022. The influent samples were collected and compared over a period of nine study weeks per year. The Wilcoxon signed rank test was used to assess the differences in the log-transformed SARS-CoV-2 RNA viral load in the study weeks during the Easter period in 2021 and 2022. Table 1 shows the median differences paired data of Easter period in 2021 and Easter period in 2022. All study weeks over the Easter period in 2021 had lower SARS-CoV-2 RNA viral loads than the study weeks over the Easter period in 2022. There was no significant difference in the log-transformed median SARS-CoV-2 viral load in study weeks 1 and 4 of Easter period in 2021 and 2022.

The SARS-CoV-2 viral load in wastewater during the Easter week was compared between 2021 and 2022. The SARS-CoV-2 viral load in both Easter weeks was split into quartiles and assigned colors as shown on the legend in Figure 1. Blue represented very low viral load, green represented low viral loads, yellow represented medium viral loads, orange represented high viral loads and red represented very high viral loads. About 57% of the WWTPs were observed to be in the high viral loads category over the Easter week in 2022. Approximately 52% of the WWTPs moved from a lower category over the Easter week in 2021 to a higher category over the Easter week in 2022. Green Point and Camps Bay WWTPs were in the red category over the Easter week in 2021 and 2022, respectively. Green Point and Camps Bay are popular leisure districts in the City of Cape Town. The log-transformed median SARS-CoV-2 RNA viral load observed on the Easter week in 2021 and Easter week in 2022 were 13.1 and 13.6 gc/day/100,000 inhabitants, respectively. There was a statistically significant difference in the log-transformed median SARS-CoV-2 viral load observed during the Easter week in 2021 and Easter week in 2022 (*p*-value = 0.0052).

Figure 2 shows SARS-CoV-2 RNA viral load between Easter week and non-Easter weeks over the Easter period in 2021 and 2022. The Easter week in 2021 and 2022 was indicated by a red dotted line. The non-Easter weeks represented study weeks before and after the Easter week (Easter Monday). No statistically significant difference in the SARS-CoV-2 RNA viral load was observed in the Easter week and study week 4, 5 and 9 (non-Easter weeks) over the Easter period in 2021 (Figure 2A). The highest log-transformed median SARS-CoV-2 RNA viral load was recorded in the Easter week (study week 6) over the Easter period in 2021. A fluctuation in the RNA signal (study week 1 to study week 8) was observed over the Easter period in 2022. There was no statistically significant difference in the SARS-CoV-2 RNA viral load in the non-Easter weeks (study week 2, 3, 4 and 6) and Easter week (Figure 2B). The highest log-transformed median SARS-CoV-2 viral load in wastewater was observed on study week 5, just after the national holiday on study week 4 over the Easter period in 2022.

We used mixed-effects modeling to observe the relationship between the changes in log-transformed SARS-CoV-2 RNA viral load on the Easter week and non-Easter weeks. The changes in the log-transformed SARS-CoV-2 RNA viral load was found to be associated with the Easter week in 2021 (*p* = 0.0019). There was no association between changes in log-transformed SARS-CoV-2 RNA viral load in Easter week and non-Easter weeks during the Easter period in 2022 (*p* = 0.7536). We also used a mixed-effect model to assess the relationship between area (Northern and Southern suburbs) at which the WWTPs are located and SARS-CoV-2 RNA viral loads. There was no significant association observed between the area and SARS-CoV-2 RNA viral loads for both Easter periods (*p* = 0.0820 and *p* = 0.6892 in 2021 and 2022, respectively).

The COVID-19 confirmed cases during the Easter period in 2021 (Figure 3A) remained steady with a slight increase in study week 7 and 8, and the percentage test positivity rate was relatively low (<5%) (Figure 3B). On the other hand, the confirmed cases during the Easter period in 2022 showed an uptick after the Easter week in 2022 (study week 8), while the test positivity rate exhibited a sharp increase after study week 7 from 17% to 28%.

## 4. Discussion

The second COVID-19 wave in South Africa, which peaked in January 2021, was associated with the Beta (B.1.1351) variant which was the predominant variant circulating during the Easter period in 2021 [3]. The Omicron (B.1.1. 529) variant was the predominant variant, which drove the fourth COVID-19 wave in late November 2021 and peaked in December 2021 [32,33]. The Omicron variant was still the predominant variant in Easter 2022 [6]. Previous studies have reported that the Omicron variant had the highest transmission rate compared to previous variants [4,34,35,36]. Reports also show that the Omicron variant rapidly spread across the globe over a short period of time. Our results showed that the COVID-19 cases and test positivity rates were higher over the Easter period in 2022 than in the Easter period in 2021. Our findings also showed a higher SARS-CoV-2 RNA signal in wastewater over the Easter period in 2022 than in the Easter period in 2021.

Wastewater surveillance of SARS-CoV-2 has assisted public health and government officials in recommending interventions towards the spread of the COVID-19 in communities. The wastewater-based epidemiology approach captures symptomatic, pre-symptomatic and asymptomatic cases. During the Easter period in 2021, the lockdown regulations under the National State of Disaster in response to COVID-19 pandemic in South Africa, limited the number of people in religious and social gatherings (100 people indoors and 250 people outdoors). The Easter period in the Western Cape in 2021 occurred when the province had just exited the second COVID-19 wave in late January 2021 and was about to enter the third COVID-19 wave in May 2021. An increase in RNA signals in wastewater over the Easter week in 2021 (study week 6) pointed to an increase in social gatherings and a slight increase in confirmed cases in study week 7 alluded to the increase COVID-19 infections. Previous studies also reported that the spike in the RNA signal in wastewater corresponded to increases in COVID-19 infections during the holiday period [37,38]. A significant change in the SARS-CoV-2 RNA viral load over the Easter week in 2021 also indicated an increase in social gatherings. 

The Western Cape province exited the fourth COVID-19 wave in late January 2022. During the Easter period in 2022, South Africa’s National State of Disaster was terminated (4th of April 2022—study week 6) and the restrictions on social gatherings and travelling were removed. According to the wastewater data over the Easter period in 2022, an increase in mobility of people in the City Cape Town may have resulted in the increase in transmission rates of COVID-19 as evident from the spike in the viral load in wastewater after study week 4. The clinical data on the other hand showed a spike in the COVID-19 infections in study week 9. These findings are consistent with the other research where a correlation was observed between the trends of mobility and transmission of SARS-CoV-2 [39,40]. The wastewater surveillance data captured the significant change in the mobility of people in the City of Cape Town and spread of COVID-19 during the Easter holidays. The influx of tourists and locals into the holiday destinations such as Green Point and Camps Bay may have caused the significantly higher SARS-CoV-2 RNA signal in wastewater over the Easter week in 2021 and 2022, respectively. Green Point and Camps Bay host various activities and historic attractions. 

A limitation to our study was that the SARS-CoV-2 RNA concentration was normalized by the design capacity of the WWTPs instead of the flow rate. Therefore, we used the population size data to further normalize the SARS-CoV-2 viral load to compare RNA viral loads across WWTPs. The study highlights the importance of mitigation strategies as intervention to minimize the spread of COVID-19 during the holiday period.

## 5. Conclusions

The present study showed that the Omicron variant was more infectious than the Beta variant with higher number of COVID-19 cases and SARS-CoV-2 RNA viral load detected in wastewater during the Easter period in 2022. The median SARS-CoV-2 RNA viral loads were significantly higher over the Easter week in 2022 compared to the Easter week in 2021. Popular holiday destinations, Green Point and Camps Bay WWTPs had highest SARS-CoV-2 RNA viral load over the Easter week in 2021 and 2022, respectively. The Easter week in 2021 had an impact on the SARS-CoV-2 viral load over the Easter period in 2021. Strategies must be put in place to mitigate the spread of the infectious virus during the holiday seasons including targeted public health campaigns.

## Figures and Tables

**Figure 1 viruses-15-00162-f001:**
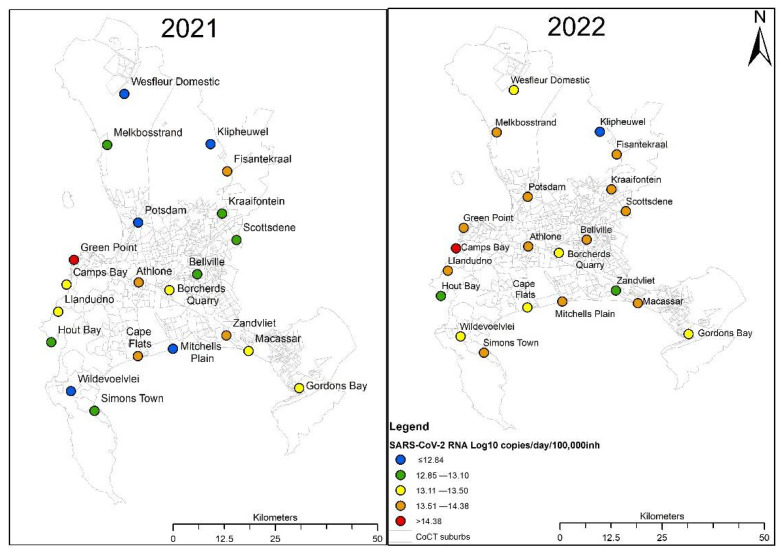
SARS-CoV-2 RNA viral load during the Easter week in 2021 and Easter week in 2022.

**Figure 2 viruses-15-00162-f002:**
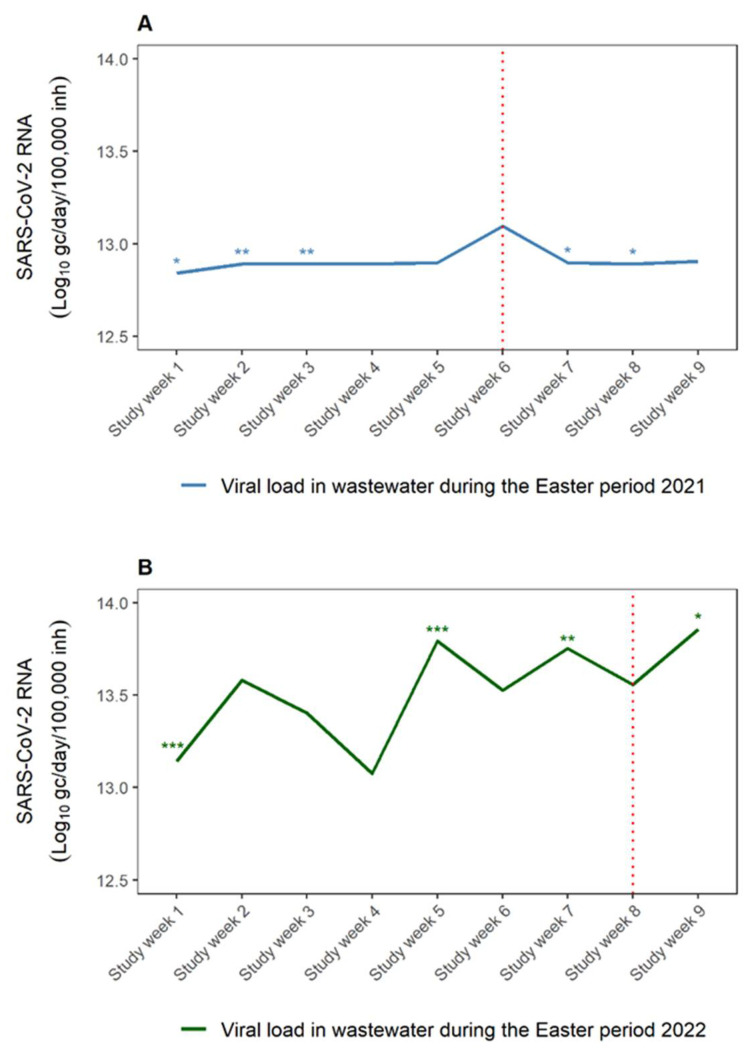
SARS-CoV-2 RNA viral load in wastewater during the (**A**) Easter period in 2021 and (**B**) Easter period in 2022. Red vertical dotted line indicates the Easter Monday in 2021 and 2022. * *p* < 0.05, ** *p* < 0.01, and *** *p* < 0.001 viral load compared between Easter week (Easter Monday) and non-Easter weeks.

**Figure 3 viruses-15-00162-f003:**
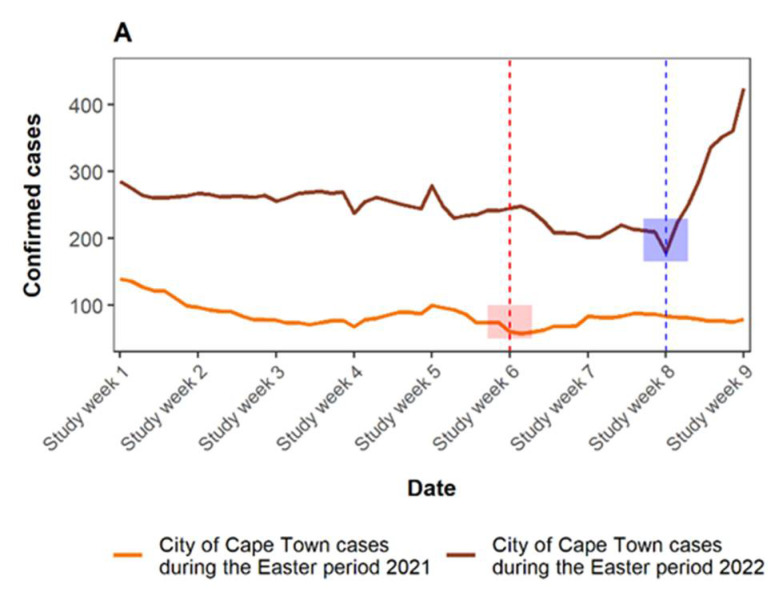
COVID-19 confirmed cases (**A**) percent positivity test rate (**B**) in the City of Cape Town during the Easter period in 2021 and Easter period in 2022. Red and blue vertical dashed line and boxes indicates the Easter Monday in 2021 and 2022, respectively.

**Table 1 viruses-15-00162-t001:** Paired comparison of log-transformed median SARS-CoV-2 viral load in wastewater in each study week during the Easter period in 2021 and Easter period in 2022.

	Dates	Paired Differences (Log_10_ gc/Day/100,000 inh)	Standard Deviation	*p*-Value
Study week 1	1 March 2021 and 28 February 2022	−0.1667	0.5001	0.1424
Study week 2	8 March 2021 and 7 March 2022	−0.6215	0.6104	<0.001
Study week 3	15 March 2021 and 14 March 2022	−0.5486	0.4373	<0.001
Study week 4	22 March 2021 and 21 March 2022	−0.4350	0.5211	0.1936
Study week 5	29 March 2021 and 28 March 2022	−0.9105	0.5522	<0.001
Study week 6	5 April 2021 and 4 April 2022	−0.3168	0.6036	0.0345
Study week 7	12 April 2021 and 11 April 2022	−0.8657	0.5038	<0.001
Study week 8	19 April 2021 and 18 April 2022	−0.5838	0.4106	<0.001
Study week 9	26 April 2021 and 25 April 2022	−0.5920	0.6052	<0.001

## Data Availability

Data available from author upon reasonable request.

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
