# Peer review of "Using Wastewater Surveillance to Compare COVID-19 Outbreaks during the Easter Holidays over a 2-Year Period in Cape Town, South Africa"

_viruses, 2023, doi:10.3390/v15010162_

Round 1

Reviewer 1 Report

The works bears merit. Can be published with minor modifications and improvements. More review on relevant work. More references to be added. 

Author Response

Response to Reviewer 1 Comments

Point 1: English language and style

( ) English very difficult to understand/incomprehensible
( ) Extensive editing of English language and style required
( ) Moderate English changes required
(x) English language and style are fine/minor spell check required
( ) I don't feel qualified to judge about the English language and style

Author Response: Thank you for your comment. We have conducted a spell check as suggested.

Point 2:

Yes

Can be improved

Must be improved

Not applicable

Does the introduction provide sufficient background and include all relevant references?

( )

(x)

( )

( )

Are all the cited references relevant to the research?

( )

(x)

( )

( )

Is the research design appropriate?

(x)

( )

( )

( )

Are the methods adequately described?

(x)

( )

( )

( )

Are the results clearly presented?

( )

(x)

( )

( )

Are the conclusions supported by the results?

( )

(x)

( )

( )

Author Response: Thank you for your comments. We have improved the introduction (please see page 2, paragraph 2 lines 63-75), discussion (please see page 7 & 8, paragraph 1 lines 237-238, paragraph 2 lines 254-255 & paragraph 3 lines 263-270) sections and cited references relevant to the research.

Point 3: Comments and Suggestions for Author: The works bears merit. Can be published with minor modifications and improvements. More review on relevant work. More references to be added. 

Author Response: Thank you for your comment. We have included reviewed relevant work and included more references as suggested.

Reviewer 2 Report

Review report

Article: „Using wastewater surveillance to compare COVID-19 outbreaks during the Easter holidays over a 2-year period in Cape Town, South Africa“(Manuscript ID: viruses-2115494)

The authors outlined the importance of wastewater surveillance for determination of early infection outbreaks. They monitored SARS-CoV-2 RNA levels in wastewater during the period of nine weeks during Easter holidays (holiday gathering and travelling have played a significant role in the transmission of COVID-19). Their findings showed that there is a statistically significant difference in viral load which, once again, highlighted the wastewater surveillance as an important tool for early infection outbreaks determination.  

Introduction - The authors have stated the objectives, aim of the research and explained the relevance of the presented research.

In 2021, many countries have employed a recommendation to put in place wastewater monitoring to track COVID-19 and its variants, which has contributed to the early detection of the virus and its variants. The results of this work provide an advancement of the current knowledge for the region of Cape Town which is one of the leading COVID-19 hotspots in the Western Cape province, South Africa.

Materials and Methods - The authors described the procedure of collecting the wastewater samples, procedure for viral concentration, RNA isolation and qRT-PCR. 

The methods and reagents are described with sufficient details to allow another researcher to reproduce the results.

Results – The authors collected the total of 354 samples on Easter holidays in 2021 and 2022. They used Wilcoxon signed rank test to assess the differences in the viral load and mixed-effects modeling to observe the relationship between the changes in viral load on the Easter week and non-Easter weeks and the relationship between area. 

The figures and tables are appropriate, they show the data properly and they are easy to interpret and understand.  

Discussion and Conclusions - The conclusions are consistent with the evidence and arguments presented.

The conclusions are interesting for the readership of the journal. Since the results and conclusions are written clearly and understandably, the paper will attract a wide readership.

Author Response

Response to Reviewer 2 Comments

Point 1: English language and style

( ) English very difficult to understand/incomprehensible
( ) Extensive editing of English language and style required
( ) Moderate English changes required
(x) English language and style are fine/minor spell check required
( ) I don't feel qualified to judge about the English language and style

Author Response: Thank you for your comment. We have conducted a spell check as suggested.

Point 2:

Yes

Can be improved

Must be improved

Not applicable

Does the introduction provide sufficient background and include all relevant references?

(x)

( )

( )

( )

Are all the cited references relevant to the research?

(x)

( )

( )

( )

Is the research design appropriate?

(x)

( )

( )

( )

Are the methods adequately described?

(x)

( )

( )

( )

Are the results clearly presented?

(x)

( )

( )

( )

Are the conclusions supported by the results?

(x)

( )

( )

( )

Author Response: Thank you for your comments

Point 3: Article: „Using wastewater surveillance to compare COVID-19 outbreaks during the Easter holidays over a 2-year period in Cape Town, South Africa“(Manuscript ID: viruses-2115494)

The authors outlined the importance of wastewater surveillance for determination of early infection outbreaks. They monitored SARS-CoV-2 RNA levels in wastewater during the period of nine weeks during Easter holidays (holiday gathering and travelling have played a significant role in the transmission of COVID-19). Their findings showed that there is a statistically significant difference in viral load which, once again, highlighted the wastewater surveillance as an important tool for early infection outbreaks determination. 

Introduction - The authors have stated the objectives, aim of the research and explained the relevance of the presented research.

In 2021, many countries have employed a recommendation to put in place wastewater monitoring to track COVID-19 and its variants, which has contributed to the early detection of the virus and its variants. The results of this work provide an advancement of the current knowledge for the region of Cape Town which is one of the leading COVID-19 hotspots in the Western Cape province, South Africa.

Materials and Methods - The authors described the procedure of collecting the wastewater samples, procedure for viral concentration, RNA isolation and qRT-PCR. 

The methods and reagents are described with sufficient details to allow another researcher to reproduce the results.

Results – The authors collected the total of 354 samples on Easter holidays in 2021 and 2022. They used Wilcoxon signed rank test to assess the differences in the viral load and mixed-effects modeling to observe the relationship between the changes in viral load on the Easter week and non-Easter weeks and the relationship between area. 

The figures and tables are appropriate, they show the data properly and they are easy to interpret and understand. 

Discussion and Conclusions - The conclusions are consistent with the evidence and arguments presented.

The conclusions are interesting for the readership of the journal. Since the results and conclusions are written clearly and understandably, the paper will attract a wide readership.

Author Response: We thank the reviewer for the comments.